# The Role of Kidney Biopsy in Fabry Disease

**DOI:** 10.3390/biomedicines13040767

**Published:** 2025-03-21

**Authors:** Irene Capelli, Laura Martano, Gian Marco Berti, Gisella Vischini, Sarah Lerario, Vincenzo Donadio, Alex Incensi, Valeria Aiello, Francesca Ciurli, Benedetta Fabbrizio, Stefano Chilotti, Renzo Mignani, Gianandrea Pasquinelli, Gaetano La Manna

**Affiliations:** 1Nephrology, Dialysis and Renal Transplant Unit, IRCCS Azienda Ospedaliero-Universitaria di Bologna, 40138 Bologna, Italy; 2Department of Medical and Surgical Sciences (DIMEC), Alma Mater Studiorum University of Bologna, 40126 Bologna, Italygianandr.pasquinelli@unibo.it (G.P.); 3Neuromuscular and Neuroimmunology Unit, Bellaria Hospital, IRCCS Institute of Neurological Sciences of Bologna, 40139 Bologna, Italy; vincenzo.donadio@unibo.it (V.D.);; 4Department of Biomedical and Neuromotor Sciences (DIBINEM), Alma Mater Studiorum University of Bologna, 40126 Bologna, Italy; 5Pathology Unit, IRCCS Azienda Ospedaliero-Universitaria di Bologna, 40138 Bologna, Italystefano.chillotti@aosp.bo.it (S.C.)

**Keywords:** biopsy, histopathology, Fabry nephropathy, podocyte, inclusions

## Abstract

Fabry disease (FD) is a rare X-linked lysosomal storage disorder caused by mutations in the GLA gene, leading to α-galactosidase A deficiency and subsequent accumulation of glycosphingolipids, including globotriaosylceramide (Gb3) and globotriaosylsphingosine (lyso-Gb3), in multiple organs. This accumulation can result in multisystemic disease and life-threatening complications. FD presents with a broad phenotypic spectrum, ranging from the classic form, with early and severe symptoms, to a later-onset form with variable manifestations. The severity of the disease in females is more variable due to X-chromosome inactivation (XCI). Renal involvement is a key feature, and kidney biopsy remains a valuable tool for diagnosing FD and assessing the extent of nephropathy. Although molecular genetic testing is the gold standard for diagnosis, kidney biopsy aids in confirming renal involvement, detecting coexisting conditions, and determining the pathogenicity of variants of uncertain significance (VUSs). Moreover, kidney biopsy can serve as a prognostic tool by identifying early markers of nephropathy, such as foot process effacement and glomerular sclerosis, which predict disease progression. Emerging technologies, including machine learning, offer the potential to enhance the analysis of renal histology, improving diagnostic accuracy and patient stratification. Despite the challenges posed by overlapping diseases and potential misdiagnoses, kidney biopsy remains an essential component of FD diagnosis and management, facilitating early detection, the monitoring of disease progression, and the evaluation of therapeutic responses.

## 1. Introduction

Fabry disease (FD, OMIM 301500) is a rare X-linked lysosomal storage disorder caused by pathogenic variants in the GLA gene encoding the enzyme α galactosidase A (α-Gal A) [1].

The lack or deficiency of α-Gal A leads to a progressive accumulation of glycosphingolipids, mainly globotriaosylceramide (Gb3) and its derivative globotriaosylsphingosine (lyso-Gb3) in plasma and in a wide range of cells throughout the body, including in the kidney, heart, and central and peripheral nervous systems [2]; this accumulation can cause multisystem disease, life-threatening complications, and an increased risk of premature death [1].

The disease exhibits a broad phenotypic spectrum with the expression largely influenced by the type of GLA gene mutation that can be classified into classic or non-classic phenotypes and the patient’s sex [3].

The classic phenotype is characterized by severely reduced or absent α-Gal A activity, with symptoms such as neuropathic pain, cornea verticillata, gastrointestinal symptoms, hypohidrosis, and angiokeratoma appearing from early childhood [1,2,3].

In contrast, the late-onset phenotype presents varying levels of residual α-Gal A activity, age of onset, and manifestations, which may be mild or limited to single-organ involvement and occur later in life [2].

Phenotypes are well defined in males, while the severity of the disease in females is more variable due to X-chromosome inactivation (XCI), also known as lyonization [1].

Regarding renal involvement, the commonly used clinical markers for the diagnosis and monitoring of nephropathy secondary to Fabry disease are proteinuria, estimated glomerular filtration rate (eGFR) value, and eGFR slope. Usually, it progresses alongside other organ manifestations, making it a valuable surrogate marker for disease monitoring. Kidney damage begins early, with Gb3 accumulation in podocytes and vascular endothelium preceding overt clinical signs such as proteinuria and declining eGFR. As the disease advances, renal impairment often parallels cardiac hypertrophy and cerebrovascular involvement, with proteinuria. In later stages, accelerated kidney dysfunction coincides with heart failure and severe neurological complications [4]. However, in some cases of classic forms of FD in adults, signs of renal involvement may be absent, even though a renal biopsy reveals typical and diffuse lesions of Fabry nephropathy, such as zebra bodies [5]. In these cases, histological findings precede the onset of renal dysfunction, thereby contributing to a more precise definition of the diagnostic process.

As in many nephropathies, renal histology can still play an important role today in the diagnosis of the disease, in the assessment of its progression, and in the definition of pathogenicity of variants of unknown significance (VUS). Furthermore, there are intriguing prospects for the future application/research of machine learning techniques in the analysis of renal histology.

## 2. The Kidney Biopsy in the Diagnosis of Fabry Disease

### 2.1. Algoritm of Diagnosis of Fabry Disease

In patients with suspected FD based on clinical findings, diagnosis is confirmed through biochemical testing (enzymatic activity in males) and genetics testing (in females and males with reduced enzymatic activity).

Although renal biopsy does not represent the absolute “gold standard” for the diagnosis of Fabry disease (which is still represented by molecular genotype analysis), it nevertheless constitutes a very important diagnostic tool for identifying or confirming renal involvement in the disease. In particular, the purpose of renal biopsy is to confirm the clinical suspicion of renal damage and to verify the possible coexistence of other alterations (cases of ’overlapping’ typical patterns are described in the literature) (Figure 1) [6,7].

Biopsy is needed together with other clinical, instrumental, and laboratory tests to demonstrate the presence of pathologic issues in variants of uncertain significance or in polymorphism [1,2,8,9,10]. Rarely, genetics tests may be negative instead of raising high clinical suspicion; in these cases, it is important to find the characteristic lysosomal inclusions in the affected organ (e.g., heart, kidney, or other than skin) through biopsies.

In late-onset cases for males and females, kidney biopsy may be needed to evaluate the disease burden.

Indeed, the consensus proposed by the Anderson-Fabry Disease Italian Board in 2015 extensively described all additional information regarding the severity of underlying tissue injury that a kidney biopsy could provide: the extent of GL-3 accumulation in the various cellular elements; chronicity status; confirmation or exclusion of overlapping glomerular diseases, which may require different therapeutic choices [11]; and response to treatment [7].

### 2.2. Diagnostic Finding on Kidney Biopsy in FD

Nowadays, the procedure is a part of the clinical practice routine and uncomplicated most of the time [12].

The morphological changes in FD patients, marked by Gb3 cell deposits, are detectable in renal biopsy samples, even in early-onset cases in children [13].

Renal biopsy specimens are analyzed using three techniques: light microscopy (LM), immunofluorescence (IF), and transmission electron microscopy (EM).

In light microscopy, the biopsy is fixed in formalin or Serra’s fixative, and after processing, the 2–3 µm thick paraffin sections are commonly stained with hematoxylin and eosin (H&E) and periodic acid-Schiff (PAS). This allows for the observation of clear, empty vacuoles, appearing as transparent bubbles in the cytoplasm of the involved cell types. (Figure 2).

Gb3 deposits are present in all four kidney compartments: vascular, glomerular, tubular, and interstitial. Podocytes are the most affected cells, and quantification of the empty vacuoles is necessary [14,15]. Indeed, initially, the disease appears in glomerular podocytes with hypertrophic podocytes, and these deposits become more pronounced as the disease advances, spreading to other renal cells. In severe cases, the mesangial matrix may expand, leading to segmental or global glomerulosclerosis, interstitial fibrosis, tubular atrophy, and arteriosclerosis [15,16].

The vacuoles, due to their staining characteristics, can be identified in podocytes and epithelial cells of the distal tubules. However, they may go unnoticed when they are small and few in number, such as in the endothelial cells of arterioles or peritubular capillaries. Therefore, exclusive analysis with light microscopy may underestimate the extent of disease involvement in the various compartments of the renal cortex.

The progression of kidney fibrosis in Fabry disease is not as well defined as in diabetic nephropathy, but emerging evidence suggests a similar pattern [17]. This includes early podocyte injury and fibrosis generated by epithelial cells, which increases as the disease progresses. Early reports of Fabry nephropathy noted a significantly thickened glomerular basement membrane (GBM), along with GBM duplications and expansion of the glomerular mesangial extracellular matrix (ECM) [13,15,17]. Glomerulosclerosis and interstitial fibrosis are already present in children with early-stage tissue injury, characterized by preserved renal function and microalbuminuria (<300 mg/g), along with features of podocyte injury such as segmental foot process effacement [13]. Glomerular sclerosis and interstitial fibrosis may also be observed in females with normal renal function and in the absence of significance proteinuria (<300 mg/day) [17,18].

FD patients lack immune complex deposits, with test results either negative or non-specific, sometimes showing IgM or C3 in sclerotic areas [15,16].

Gb3 deposits are detectable both in urine and in kidney biopsy. In urine sampling, Gb3 deposits exhibit birefringence, creating a distinctive Maltese cross pattern under polarized light, and they naturally fluoresce under ultraviolet light [16]. In kidney biopsy, immunohistochemistry assay on frozen tissue may detect Gb3 deposits.

By using plastic-embedded 0.5 µm thick sections stained with toluidine blue, light microscopy shows that the empty vacuoles appear as intense dark blue inclusions, allowing for a more accurate documentation of renal parenchyma involvement.

Electron microscopy, performed on plastic-embedded sections with a thickness of 60–80 nm, reveals that these deposits consist of lysosomes containing whorls of electron-dense lamellae alternating with electron-lucent spaces. This characteristic ultrastructural pattern is highly specific and aids in confirming the diagnosis.

Depending on the orientation of the cutting plane relative to the spatial configuration of the inclusion, digital images are produced that have been described by different observers as “myelin figures”, “onion skin”, or “zebra bodies” (Figure 3) [15,19]. It is noteworthy that in semithin sections stained with toluidine blue, Gb3 accumulation is clearly detectable in all kidney compartments, and this is considerably timesaving compared to ultrastructural studies (Figure 4).

In skin biopsies, similarly, Gb3 deposits characteristic of patients with classical GLA mutations can be identified. These deposits are linked to diminished skin innervation, indicating potential indirect damage to peripheral small nerve fibers [20].

### 2.3. The Histological Scoring

The International Study Group of Fabry Nephropathy (ISGFN) has proposed a standardized scoring system for assessing histological involvement in Fabry nephropathy [21]. Pathological findings such as fibrosis, sclerosis, vascular changes, and GL-3 deposits are scored and included in the final scoring sheet to identify common features in the biopsies. The semi-quantitative approach is applied. Histological findings are evaluated as present or absent or graded by severity: none (0), mild (1), moderate (2), or severe (3). This elegant classification system encompasses light microscopy analysis both on conventional paraffin sections and semithin sections of plastic-embedded toluidine blue stain.

This is the largest histology-based study of Fabry nephropathy, examining 59 cases (35 males and 24 females) which were scored independently of clinical information. Each patient had a confirmed classic mutation except for five male patients exhibiting the p.N215S defined as late onset [22]. The standardized scoring system showed that even in the early/baseline biopsy evaluation displaying mild clinical presentation, disease-specific lesions were present [21].

### 2.4. The Overlapping Diseases and Misdiagnosis

However, it is crucial not to rely solely on the histological report, as overlapping features may suggest different diseases [6]. There is a risk of misdiagnosing a patient with Fabry disease based on electron microscopy findings alone that reveal myeloid bodies. These structures, which may be difficult to distinguish from specific Fabry-related inclusions, can result from prolonged fixation, the use of amphiphilic drugs, as well as other genetic disorders. As a result, they may be found in kidney biopsies of patients without any clinical evidence of Fabry disease.

The coexistence of Fabry disease (FD) with other immune diseases such as rheumatoid arthritis, systemic lupus erythematosus [23], and celiac sprue has been documented in the literature. FD has also been associated with various nephropathies, including membranous nephropathy [24], minimal change disease (MCD), pauci-immune glomerulonephritis [25], and IgM and IgA nephropathy (IgAN) [26,27,28,29].

However, a specific link between glomerulopathies and FD has not yet been established.

The presence of both glomerulopathies and FD can accelerate the decline in renal function, and the combination of these diseases in young patients should prompt consideration of differential diagnoses.

The glomerulopathies most commonly associated with FD are IgAN and pauci-immune glomerulonephritis [25,27].

It has been suggested that glycosphingolipids accumulated in FD continuously and chronically stimulate the immune system and induce an autoimmune reaction [25,27]. Indeed, high levels of anti-IgA1 autoantibodies have been found in patients with FD. The glycolipid structure of globotriaosylceramide resembles nephritogenoside, which causes progressive experimental nephritis similar to IgAN. Thus, some patients with FD may be prone to developing IgAN, influenced by different genetic and/or environmental factors [27].

In these cases, a kidney biopsy is essential to evaluate histological findings. Histologically, patients with IgAN and FD may show mesangial matrix widening, mesangial cell proliferation, and marked enlargement of podocytes with prominent lacy and clear cytoplasm on hematoxylin and eosin staining under light microscopy. Immunofluorescence staining reveals diffuse immunoglobulin A deposits in the mesangium. In advanced cases when glomerular sclerosis is predominant, myelin bodies may be invisible under light microscopy. In such cases, electron microscopy can aid in identifying myelin-like figures in the cytoplasm of podocytes [27,29]. The presence of diffuse crescentic glomerulonephritis on light microscopy, on the other hand, with less or no immune deposits in immunofluorescence studies, suggests pauci-immune glomerulonephritis [25].

The presence of zebra bodies in histological patterns is closely linked to other cases that may indicate a diagnosis of FD. In some instances, a normal evaluation of enzymes or genotypes has resulted in a misdiagnosis of FD. Consequently, the diagnostic algorithm should take into account the exclusion of histological findings associated with drug-induced phospholipidosis (DIP), toxins, and other hereditary conditions such as nail–patella syndrome and Niemann–Pick disease [30,31,32,33,34].

Many drugs can induce iatrogenic phospholipidosis, and the most common are amiodarone, chloroquine, carbamazepine, ranolazine, contrast medium, and hydroxychloroquine. The pathological features of iatrogenic phospholipidosis can often be indistinguishable from those of FD in renal tissue studies. However, in many cases, the involvement is primarily limited to podocytes. In the case of gentamicin, an aminoglycoside antibiotic, the primary effect is on tubular cells rather than podocytes [33,34,35,36].

Nail–patella syndrome (NPS; OMIM #161200) is an autosomal dominant disorder caused by variants in the LMX1B gene. It is classically characterized by skeletal abnormalities and extra-skeletal manifestations including ocular, renal, and neurological issues. Renal involvement occurs in 30–50% of cases, with hematuria, proteinuria, and end-stage renal disease (ESRD) in about 5% of patients. Differentiating NPS from FD is usually possible due to clinical symptoms. However, in cases with limited renal involvement, known as nail–patella-like renal disease (NPLRD) or LMX1B-associated nephropathy (OMIM #256020), the diagnosis can be more challenging [31].

Electron microscopy findings exhibit a spectrum of abnormalities, from negligible changes to the presence of electron-lucent regions in thickened GBM characterized by a “moth-eaten” appearance. Notably, there is a distinctive deposition of curvilinear type III collagen fibrils within the GBM, as well as, to a lesser extent, in the mesangium and tubular basement membranes (Table 1) [34].

Recently in the literature, some studies have shown cases of renal involvement by Niemann–Pick disease types A/B and C, an autosomal recessive storage disease caused by variants in the NPC1 or NPC2 gene, which is manifested by abnormal cholesterol and glycosphingolipid accumulation in lysosomes of various cells [32].

Renal involvement is rare but can manifest with hematuria, proteinuria, and progressive chronic renal impairment. Kidney biopsy findings include foamy podocyte cytoplasm observed under light microscopy and the presence of lamellar bodies within podocytes as seen in electron microscopy (Table 1) [32].

In these cases, a careful ultrastructural assessment is necessary because in Fabry disease, myeloid bodies are widespread in all renal cellular compartments, in contrast to the prevalence of smaller and podocyte-limited lamellar bodies in non-Fabry kidney biopsies and their configuration [32,34].

### 2.5. The Use of Kidney Biopsy in the Definition of VUS Pathogenicity and in the Female Patients

Sex and complicated genetic backgrounds, such as GLA variants of unknown significance (VUSs) confer variability in the clinical and histological picture of the disease [2,37,38]. Due to X-chromosome inactivation, enzyme activity assays may not always provide reliable information for females with FD [39]. Therefore, the primary challenge for physicians in this field is trying to understand and interpret the pathogenicity of the variant or to address the therapeutic choice in the heterozygous Fabry disease females. In these cases, tissue analysis (kidney biopsy) could help to recognize the phenotypic relevance of the VUS and the female mosaic [38].

Indeed, in these mutations defined as VUSs and debated in the scientific community, the search for Gb3 deposits in various tissues can determine the pathogenicity of the mutation. This is illustrated in the case report by Cerón-Rodríguez et al. on the c.352C>T/p.Arg118Cys variant, supporting the pathogenicity of a variant previously considered non-pathogenic [40], and in the case report by Lau et al. on the p.Ser126Gly variant, where the absence of deposits confirms the non-pathogenicity of the variant [41].

An increasing number of studies are proving that even in these complicated cases, kidney involvement is present exhibiting lipid inclusions and various degrees of vascular/glomerular sclerosis and interstitial fibrosis, confirming the critical importance of histological evaluation [42].

## 3. The Kidney Biopsy in Monitoring

The histological evaluation of FD biopsies should not be limited only to diagnosis but could be useful for longitudinal assessment of prognosis and responses to therapy; [43] however, if the significance of kidney biopsy in diagnosis is well established and widely accepted, the impact of histological findings on prognosis remains a subject of ongoing debate.

In the literature, there are some studies that evaluate possible histological prognostic findings. For example, podocyte effacement and glomerular sclerosis are two key histological markers that have been studied.

Despite proteinuria or albuminuria being considered early FD complications, glomerular lesions such as foot process effacement are early markers of nephropathy, even before laboratory data [2,13,44].

In the 2015 study by Tøndel et al., histological analysis of eight children with minimal or no albuminuria and without clear signs of nephropathy revealed that foot process effacement represents an early marker of renal damage and a prognostic indicator, especially if it appears after the initiation of enzyme replacement therapy (Table 2) [2,13,44].

Therefore, early detection of these lesions could reveal the disease in advance, allowing for the implementation of necessary measures to slow disease progression to advanced kidney disease [13].

The other prognostic marker, described by Rusu et al., is glomerular sclerosis. The patients with segmental sclerosis at kidney biopsy and abnormal levels of proteinuria at the baseline had high risk of developing progression towards kidney disease deterioration and mortality after the follow-up period (Table 2) [30].

At the same time, glomerular sclerosis predicts proteinuria, the CKD stage, and renal prognosis [30].

This monocentric study, however, presents some limitations. It demonstrates the correlation between glomerular sclerosis and worsening kidney function through a semi-quantitative analysis. However, it does not take into the account the Fogo scoring system, specifically regarding vacuolization, tubular interstitium, and vascular damage.

Evaluating early morphological parameters, even in patients with almost normal kidney function, may aid in accurately determining the patient’s prognosis and in assessing the optimal timing for initiating replacement therapy.

Nowadays, the standardized scoring system does not present a direct correlation with prognostic outcomes.

## 4. Kidney Biopsy and Evaluation of Disease Progression

Several studies have recently focused on the role of the podocyte in the development and progression of Fabry nephropathy. Since FD is storage pathology, scientific efforts have been catalyzed on the most differentiated and long-lived renal cell: the podocyte. Globotriaosylceramide (GL-3) accumulation contributes to the aging process of renal tissue [44]. Recent electron microscopy studies have analyzed podocyte damage as a driving factor in kidney injury in Fabry disease. Podocyte volume increases up to a limit beyond which the damage causes such podocyte detachment resulting in sclerosis and proteinuria. (Figure 5). It is thus conceivable that there is an inverse relationship between podocyte-GL3 volume and podocyte density. Moreover, increased podocyte foot process width (FPW) is another important marker of podocyte injury. Taken together, these indicators can be used both for the diagnosis or confirmation of the disease and for assessing the actual extent of tissue damage, as well as serving as a prognostic factor for the disease [44]. The potential of these easily obtainable markers could potentially assist clinicians in the diagnostic–therapeutic algorithm, helping them categorize patients based on their risk of progression. However, the main limiting factor is the reproducibility of the data in order to standardize the results. In this regard, some authors are investigating and developing techniques to overcome the issue. A determining factor appears to be that the variation in the Fabry podocyte phenotype within a single patient’s renal biopsy is much smaller than the variation observed between different patients, suggesting that a single glomerulus can be considered representative of this phenomenon in the biopsy. Moreover, no cross-correction between affected and non-affected podocytes is present [44].

Furthermore, Michael Mauer and Behzad Najafian’s group developed a brilliant quantification method which describes percentage of podocytes showing ultrastructural findings of globotriaosylceramide (GL-3) inclusions adjusted for age and sex [43,45,46]. These observations indicate a threshold for podocyte density below which the nephron cannot survive. Keeping podocytes’ GL3 volume below this level may help prevent critical glomerular damage (Table 2).

There is no consensus on the best method to estimate FPW, and the current gold standard is time-consuming. To address this, an automated FPW estimation technique using deep learning has been developed, which could greatly improve its applicability in daily clinical practice. This approach was recently published by the same group to accurately estimate FPW. The model was applied to electron microscopy images of kidney biopsies, and the results of FPW measured by deep learning were highly correlated with those measured by expert technicians. Given this, the substantial advantage of applying the model in terms of time efficiency, reproducibility, and accessibility in clinical practice is immediately clear [47].

These advanced studies, based on the dissection of podocytes’ ultrastructural features as the main driver of disease progression, would provide valuable insights and would lay the groundwork for precise and detailed patient stratification, enhancing the current clinical and biochemical understanding of FD disease.

Another field of research on the rise is focusing on the final step of the FD pathogenesis: fibrosis. This final convergence point is shared by numerous pathologies, even those with distinctly different etiopathogenetic mechanisms. Indeed, fibrosis is the main limiting factor of any therapeutic strategies and at the same time the main determining factor in the progression to terminal kidney damage [17]. Although only a small number of studies have investigated the pathogenesis of this process in Fabry nephropathy, it could be crucial for managing and slowing the progression of kidney damage in Fabry’s disease. Fibroblast proliferation and the production of pro-inflammatory cytokines are induced by the activation of biological processes caused by Gb3 deposition in all kinds of renal cells [17]. Targeting these mechanisms may lead to new therapeutic opportunities.

A different approach consists in identifying specific markers of the disease through the immunofluorescence technique, as described for the skin [20]. In this case, it is possible to highlight the presence of accumulations of Gb3 inside the cells of the podocytes (Figure 6). This approach allows us to obtain specific information on Fabry disease without having to resort to more expensive and complex investigations such as electron microscopy. At the moment, however, the immunofluorescence method for the analysis of accumulations of Gb3 in renal tissue is only experimental and has not been implemented in clinical practice, even if we have an ongoing trial to verify its reliability as a clinical tool.

## 5. Conclusions

Since FD is a heterogeneous disease, a multidisciplinary approach is essential to avoid diagnostic mistakes that could lead to inappropriate treatment or delays in treatment. As it appears from this literature review, scientific opinion is reinforcing the essential role of the kidney biopsy for a comprehensive FD assessment.

The unavoidable necessity to integrate clinical data (physical examination, family history, laboratory and genetic analysis) and histological findings becomes clear in order to overcome dangerous misdiagnosis.

Nowadays, kidney biopsy in Fabry disease is designated for interpreting the pathogenicity of VUSs and cases of suspected dual pathology. Recent histological studies explored also the use of kidney biopsy in the assessment of disease burden in all cases of non-classical phenotypes and in female patients.

An important role of the kidney biopsy has been suggested in the monitoring of specific therapy results over time. From a prognostic perspective, the early identification of histological markers of renal damage would also help detect patients at risk of progressing to chronic kidney disease sooner, allowing for timely intervention with personalized therapeutic decisions.

Finally, new technologies, particularly the application of artificial intelligence and deep learning, are opening innovative pathways for renal biopsy analysis.

## Figures and Tables

**Figure 1 biomedicines-13-00767-f001:**
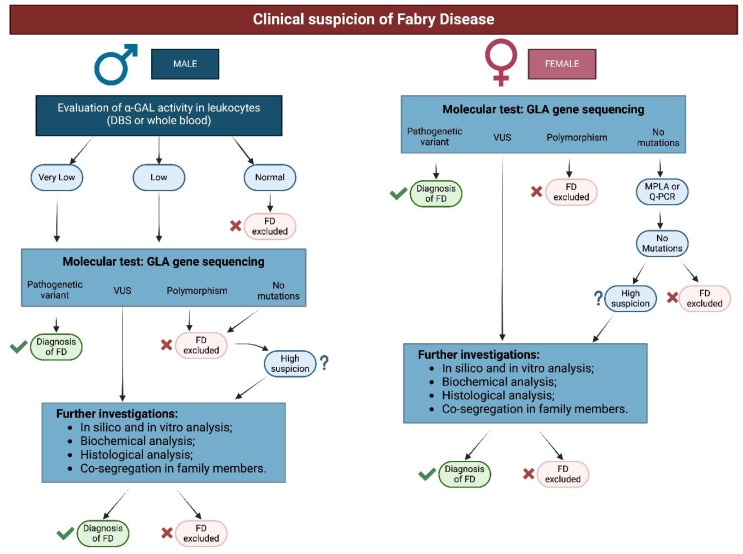
Algorithm of diagnosis of FD.

**Figure 2 biomedicines-13-00767-f002:**
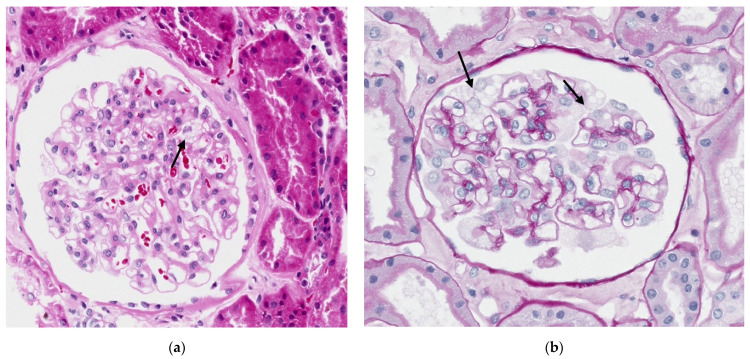
Light microscopy of the cortical renal parenchyma showing segmental vacuolated appearance of podocytes with wrinkling of the loops (indicated by arrow). (**a**) Hematoxylin and eosin (H&E ×20; scale bar 90 μm); (**b**) periodic acid-Schiff, (PAS ×20; scale bar 200 μm).

**Figure 3 biomedicines-13-00767-f003:**
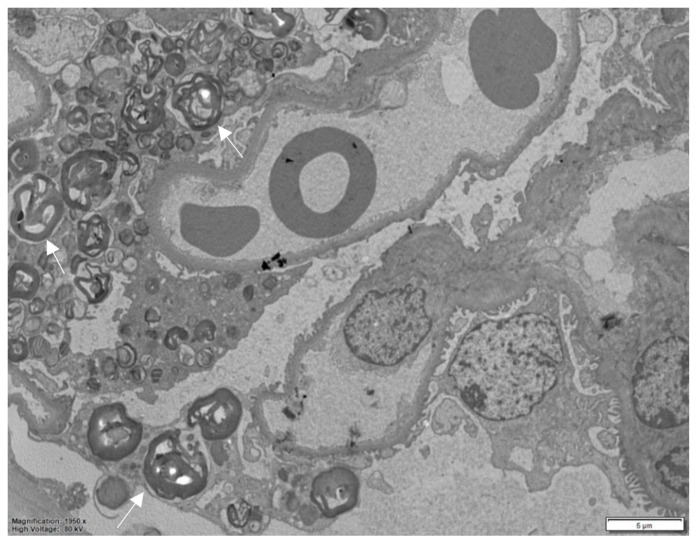
Electron microscopy, numerous myelin figures are observed within the podocyte cytoplasm in the glomerulus. The asterisk indicates an unaffected podocyte.

**Figure 4 biomedicines-13-00767-f004:**
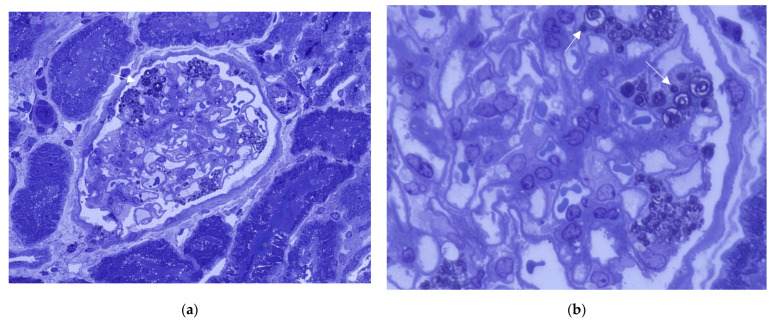
Toluidine blue stain. Abundant dense darkly stained round inclusions in podocytes of a glomerulus (indicated by arrow). (**a**) (400×); (**b**) (1000×).

**Figure 5 biomedicines-13-00767-f005:**
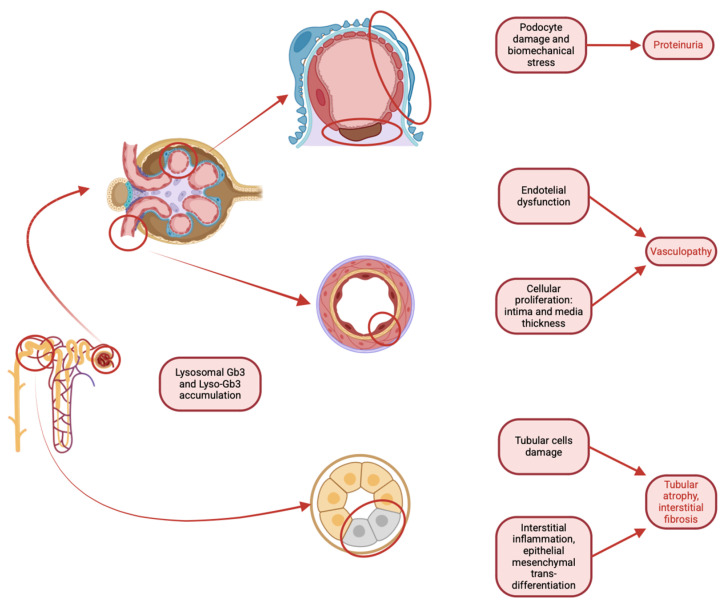
Pathological mechanisms associated with lysosomal accumulation of Gb3 and Lyso-Gb3 in the kidney, highlighting the involvement of podocytes, vascular endothelium, and tubular cells. In podocytes, the accumulation leads to an increase in volume until their detachment, resulting in proteinuria and sclerosis, while endothelial dysfunction and cellular proliferation result in vasculopathy. Tubular cell damage and interstitial inflammation contribute to tubular atrophy and interstitial fibrosis.

**Figure 6 biomedicines-13-00767-f006:**
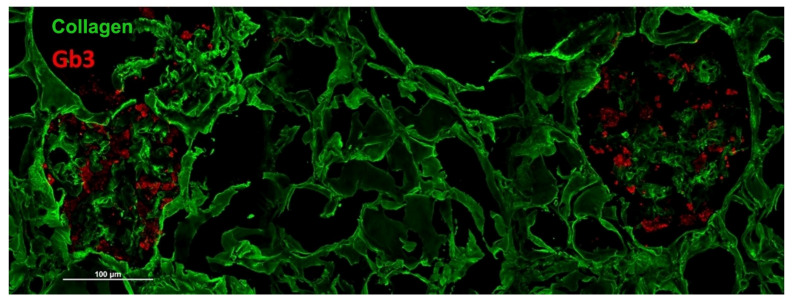
Immunofluorescence staining of Gb3 deposits (red signal) in podocytes of renal cortical parenchyma identified by a collagen staining (green signal).

**Table 1 biomedicines-13-00767-t001:** The main features and differences between FD, nail–patella syndrome, and Niemann–Pick syndrome.

	Fabry Disease	Nail–Patella Syndrome	Niemann-Pick Disease Types A/B and C
Gene	GLA	LMX1B	NPC1-NPC2
Clinical features	Neuropathic pain Hypertrophic cardiomyopathy Cornea verticillata Gastrointestinal symptoms Hypohidrosis Angiokeratoma	Skeletal abnormalities (kneecap, iliac, patellae) Nail abnormalities Ocular (glaucoma) Neurological issues	Jaundice Hepatosplenomegaly Neurological issues (supranuclear vertical gaze palsy, cerebellar ataxia) Gelastic cataplexy
Renal involvement	Hematuria Proteinuria CKD	Hematuria (30–50 cases) Proteinuria ESRD (5% of patients)	Rare Hematuria Proteinuria CKD
Kidney biopsy findings	Myeloid bodies are widespread in all renal cellular compartments	Minor modifications Moth-eaten GBM (electron-lucent regions thickened GBM) Deposition of curvilinear type III collagen fibrils on GBM	Foamy podocyte’s cytoplasm on LM lamellar bodies in podocytes on EM

**Table 2 biomedicines-13-00767-t002:** Correlation between renal histopathological findings and clinical prognosis in various studies. The table summarizes key histological markers observed in kidney biopsies and their prognostic value in disease progression.

Study/Reference	Histopathological Findings	Clinical Prognostic Correlation
Fogo et al., 2010 [21] (ISGFN)	Semi-quantitative histological scoring of fibrosis, sclerosis, vascular changes	Used for risk stratification and prognosis assessment
Najafian et al., 2011 [44]	Progressive podocyte injury and Gb3 accumulation on EM	Associated with worsening renal function and proteinuria
Weidemann et al., 2013 [17]	Renal fibrosis on LM	Marker of disease progression and resistance to therapy
Tøndel et al., 2015 [13]	Podocyte foot process effacement on EM	Early marker of nephropathy, even before proteinuria onset
Mauer et al., 2017 [43]	Reduction of Gb3 inclusion in podocyte on EM with treatment	Correlates with improved renal outcomes in treated patients
Rusu et al., 2022 [30]	Glomerular sclerosis on LM	Predicts kidney disease progression and mortality

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
