# Peer review of "The Role of Kidney Biopsy in Fabry Disease"

_biomedicines, 2025, doi:10.3390/biomedicines13040767_

Round 1

Reviewer 1 Report

Comments and Suggestions for Authors

Manuscript ID: Biomedicines-3496554

Type: Review

Title: The role of kidney biopsy in Fabry disease

General comment:

The article presents a pertinent review of kidney biopsy in Fabry disease with balanced and well-structured information. As a review-type paper, an important objective should be the teaching value for what I recommend introducing some figures and tables, reducing at the same time the length of some sections. In particular, I think the paper would gain quality and comprehension by adding a figure depicting the progressive kidney pathophysiologic changes as podocyte inclusions, modification of FPW, inflammatory activation, tubular and vascular changes, and fibrosis. Also, in the section on overlapping disorders, I recommend reducing the text by adding a table with the main similarities and differences among the different conditions.

Specific comments:

  • Abstract:
    • Line 22: “later-onset form with milder manifestations”. As you comment further below in the introduction, late(r)-onset forms are not always mild but rather present a varying degree of expression that may be severe although usually restricted to one or few organ or system affectations. It has to be differentiated from the female affectation that varies with the X-chromosomome inactivation (as indicated in lines 59-60) but may also be a classic type or late-onset presentation.
    • Line 30: “Early markers of nephropathy, such as foot process effacement and glomerular sclerosis”. I agree that FPE may represent an early marker of nephropathy but when glomerular sclerosis is present it indicates that the pathological process is already advanced to non-reversible stages.
  • Section 2.
    • Line 78: (cases of 'overlapping' typical patterns are described in the literature). Add reference(s).
    • Line 79: “to demonstrate the presence of pathologic issues in variants of uncertain significance or in benign polymorphism”. I would limit the demonstration of pathological lesions to VUS or controversial polymorphisms (perhaps not benign).
    • Line 81: in the context (instead of contest).
    • Line 83: “Other than skin”. The findings of pathologic changes in the skin biopsy have been used for diagnosis and to evaluate therapeutic efficacy. What would be the significance of finding typical inclusions in the skin biopsy for the authors?
    • Line 125: “Preserved renal function and albuminuria (<300 mg/g creatinine)”. Better microalbuminuria.
    • Line 128: “Absence of overt proteinuria”. How do you define overt proteinuria?
    • Line 194: “The glomerulopathies most commonly associated with FD are IgAN and pauci-immune glomerulonephritis”. Please, add reference(s).
    • Line 256: “The use of kidney biopsy in the definition of pathogenicity vus and in the females”. Please correct the phrase.
    • Line 262: “or the afford the therapeutic dilemma”. Correct, please.
  • Section 3.
    • Line 302: “Fogo scoring system”. Please, add a reference and describe the score.
  • Section 4.
    • Subtitle: Kidney biopsy and machine learning. Machine learning is only a new technique. I would change the subtitle to: Kidney biosy and evaluation of disease progression.
    • Line 310: “Role of the podocyte in development and progression of Fabry nephropathy”. Please, add a figure showing the different stages of Fabry nephropathy: podocyte deposits, modification of FPW, podocyte injury, glomerular damage, fibrosis, fibroblast proliferation and production of pro-inflammatory cytokines.
    • Line 311: “FD is a defective storage pathology”. Correct the phrase.
    • Line 312: “Globobotriaosylceramide (GL-3) inclusions accumulation accelerates the ageing process of renal tissue”. Please, add a reference.
    • Line 321: “as well as serving as a prognostic factor for the disease”. Please, add a reference.
  • Section 5. Conclusion
    • Line 380: “The diagnostic and prognostic significance of the biopsy procedure in various conditions of FD suggests that this investigation should be conducted more frequently. Please, be more specific and just state the most accepted indications.
  • References:
    • Please, describe references as follows: Author 1, A.B.; Author 2, C.D. Title of the article. Abbreviated Journal NameYearVolume, page range.

Author Response

Comments 1: The article presents a pertinent review of kidney biopsy in Fabry disease with balanced and well-structured information. As a review-type paper, an important objective should be the teaching value for what I recommend introducing some figures and tables, reducing at the same time the length of some sections. In particular, I think the paper would gain quality and comprehension by adding a figure depicting the progressive kidney pathophysiologic changes as podocyte inclusions, modification of FPW, inflammatory activation, tubular and vascular changes, and fibrosis. Also, in the section on overlapping disorders, I recommend reducing the text by adding a table with the main similarities and differences among the different conditions.

Response 1: We thanks the reviewer for the comments. We as suggested prepared a figure (Figure 5) to better elucidate pathophysiological issues and a table to better explain differential diagnosis features (table 1)

Comments 2: Line 22: “later-onset form with milder manifestations”. As you comment further below in the introduction, late(r)-onset forms are not always mild but rather present a varying degree of expression that may be severe although usually restricted to one or few organ or system affectations. It has to be differentiated from the female affectation that varies with the X-chromosomome inactivation (as indicated in lines 59-60) but may also be a classic type or late-onset presentation.

Response 2: We better clarify this concept as suggested changing line 22.  In particular the actual line is “FD presents with a broad phenotypic spectrum, ranging from the classic form, with early and severe symptoms, to a later-onset form with variable manifestations. The severity of the disease in females is more variable due to X chromosome inactivation (XCI)”

Comments 3: Line 30: “Early markers of nephropathy, such as foot process effacement and glomerular sclerosis”. I agree that FPE may represent an early marker of nephropathy but when glomerular sclerosis is present it indicates that the pathological process is already advanced to non-reversible stages.

Response 3: We agree with the reviewer and we updated the text as follow: “Moreover, kidney biopsy predicts disease progression by identifying early markers like foot process effacement and key prognostic indicators such as glomerular sclerosis and interstitial fibrosis, which can significantly influence the subsequent therapeutic approach”

Comments 4: Line 78: (cases of 'overlapping' typical patterns are described in the literature). Add reference(s).

Response 4: We updated as requested with reference 4 and 9

Comments 5: Line 79: “to demonstrate the presence of pathologic issues in variants of uncertain significance or in benign polymorphism”. I would limit the demonstration of pathological lesions to VUS or controversial polymorphisms (perhaps not benign).

Response 5: We changed the text as follow: Biopsy is needed together with other clinical, instrumental and laboratory tests to demonstrate the presence of pathologic issues in variants of uncertain significance or in polymorphism

Comments 6: Line 81: in the context (instead of contest).

Response 6: We corrected the sentences as requested

Comments 7: Line 83: “Other than skin”. The findings of pathologic changes in the skin biopsy have been used for diagnosis and to evaluate therapeutic efficacy. What would be the significance of finding typical inclusions in the skin biopsy for the authors?

Response 7: In this case we were illustrating skin biopsy as diagnostic tool, together with others.

Comments 8: Line 125: “Preserved renal function and albuminuria(<300 mg/g creatinine)”. Better microalbuminuria. à

Response 8: We changed the sentence as requested.

Comments 9: Line 128: “Absence of overt proteinuria”. How do you define overt proteinuria?

Response 9: >300 mg/day ref 15 So we change the sentence adding the cut-off value.

Comments 10: Line 194: “The glomerulopathies most commonly associated with FD are IgAN and pauci-immune glomerulonephritis”. Please, add reference(s).

Response 10: We updated with reference 23 and 25

Comments11: Line 256 (264): “The use of kidney biopsy in the definition of pathogenicity vus and in the females”. Please correct the phrase.

Response 11: We corrected the sentence as requested.

Comments 12: Line 262: “or the afford the therapeutic dilemma”. Correct, please.

Response 12: Therefore, the primary challenge for physicians in this field is to understand and interpret the pathogenicity of the variant or to address the therapeutic choice in heterozygous females with Fabry disease.

Comments 13: Line 302: “Fogo scoring system”. Please, add a reference and describe the score.

Response 13: We described The paragraph on the Fogo score system is at 2.3, line 162, and the related reference is 19.

Comments 14: Subtitle: Kidney biopsy and machine learning. Machine learning is only a new technique. I would change the subtitle to: Kidney biopsy and evaluation of disease progression.

Response 14: We changed as requested

Comments 15: Line 310: “Role of the podocyte in development and progression of Fabry nephropathy”. Please, add a figure showing the different stages of Fabry nephropathy: podocyte deposits, modification of FPW, podocyte injury, glomerular damage, fibrosis, fibroblast proliferation and production of pro-inflammatory cytokines

We added figure to to better elucidate this concept.

Comments 16: Line 311 (319): “FD is a defective storage pathology”. Correct the phrase.

Response 16: We Corrected as requested.

Comments 17: Line 312 (320): “Globobotriaosylceramide (GL-3) inclusions accumulation accelerates the ageing process of renal tissue”. Please, add a reference.

Response 17: We change the sentence and add the reference. Indeed Najafian et al demonstrate that GL-3 deposition in podocytes increases over time, leading to structural damage and functional decline, which are features often associated with tissue aging.

Comment s18: Line 321 (329): “as well as serving as a prognostic factor for the disease”. Please, add a reference.

Response 18: We added the reference as suggested.

Comments 19: Line 380: “The diagnostic and prognostic significance of the biopsy procedure in various conditions of FD suggests that this investigation should be conducted more frequently. Please, be more specific and just state the most accepted indications.

Response 19: We change the sentence as requested.

Nowadays kidney biopsy in Fabry disease in considered indicated in interpreting pathogenicity of VUS and in case of suspected dual pathology. Recent histological studies Explored also the use of kidney biopsy in the assessment of disease burden in all cases of non-classical phenotypes and in female patients. An important role of kidney biopsy has been suggested in the monitoring of specific therapy results over time. From a prognostic perspective, the early identification of histological markers of renal damage would also help in detect patients at risk of progressing to chronic kidney disease sooner, allowing for timely intervention with personalized therapeutic decisions. Finally, new technologies, particularly the application of artificial intelligence and deep learning, are opening innovative pathways for renal biopsy analysis.

Comments 20: Please, describe references as follows: Author 1, A.B.; Author 2, C.D. Title of the article. Abbreviated Journal NameYearVolume, page range.

Response 20: We changed as requested

Reviewer 2 Report

Comments and Suggestions for Authors

The paper is really interesting and well done. All the sections are well described. No odds for me

Author Response

Comments 1: The paper is really interesting and well done. All the sections are well described. No odds for me

Response 1: "Thank you very much for the comment.

Reviewer 3 Report

Comments and Suggestions for Authors

This is not an oryginal paper and although, as a pathologist, I always find great value in biopsies analysis, this work does not propose anything New. It is not Innovative, all the knowledge can be found in books and all manuscripts. Authors don’t share any of their own experiences. Thet not propose any novelty either diagnostic or thetapeutic. 

Comments on the Quality of English Language

English can be improved 

Author Response

Comment1: This is not an oryginal paper and although, as a pathologist, I always find great value in biopsies analysis, this work does not propose anything New. It is not Innovative, all the knowledge can be found in books and all manuscripts. Authors don’t share any of their own experiences. Thet not propose any novelty either diagnostic or thetapeutic. 

English can be improved 

Response1: We proposed a review article, that do not consent to present, together with published results also personal data. The experience of our group has been considered in the reference list as the others published.

The novelty of the proposed article is in our opinion based on the decision to summarize al the knowledge in the field from clinical and research areas.

We revised the language.

Reviewer 4 Report

Comments and Suggestions for Authors

Thank you for the opportunity to review this manuscript.

Some minor issues may improve the readability of this review:

-Add a draw scheme besides the histological findings to make it easier to understand.

-Briefly explain any clinical-histological correlation that may not be replaced with clinical or laboratory changes,  in order to highlight the importance of histologic  follow-up.

Add a table showing the clinical-histopathological prognostic correlation reported through different manuscripts, in order to help the reader determine summarise this information.

Also, briefly describe the chronological concordance of kidney involvement with other clinical milestones, in order to remark the importance of kidney involvement as a surrogate marker in the integral follow-up of this patients.

Author Response

Comments 1: Add a draw scheme besides the histological findings to make it easier to understand. 

Response 1: We prepared a figure (Figure 5) to better elucidate pathophysiological issues and histological findings as suggested

Comments 2: Briefly explain any clinical-histological correlation that may not be replaced with clinical or laboratory changes, in order to highlight the importance of histologic follow-up.

Response 2: The importance of biopsy in follow-up is in its ability to early detect signs of kidney damage, such as foot process effacement, which precedes the onset of albuminuria and proteinuria. At the same time, glomerulosclerosis, tubularinterstitial fibrosis and vessel fibrosis can provide valuable insights into the evolutionary course of renal involvement. This is particularly useful for personalizing therapeutic choices. we reported it in lines 282–284 and 289–290.

Comments 3: Add a table showing the clinical-histopathological prognostic correlation reported through different manuscripts, in order to help the reader determine summarise this information.

Response 3: We added a table as requested

Comments 4: Also, briefly describe the chronological concordance of kidney involvement with other clinical milestones, in order to remark the importance of kidney involvement as a surrogate marker in the integral follow-up of this patients.

Response 4: We added as requested

Round 2

Reviewer 3 Report

Comments and Suggestions for Authors

I Think I this papier has low scientific value and I advise against publishing review Articles 

Comments on the Quality of English Language

English could be improved